# Climate-Affected Australian Tropical Montane Cloud Forest Plants: Metabolomic Profiles, Isolated Phytochemicals, and Bioactivities

**DOI:** 10.3390/plants13071024

**Published:** 2024-04-03

**Authors:** Ngawang Gempo, Karma Yeshi, Darren Crayn, Phurpa Wangchuk

**Affiliations:** 1Australian Institute of Tropical Health and Medicine (AITHM), James Cook University, Nguma-bada Campus, McGregor Rd., Cairns, QLD 4878, Australia; ngawang.gempo@my.jcu.edu.au (N.G.); phurpa.wangchuk@jcu.edu.au (P.W.); 2College of Public Health, Medical and Veterinary Services (CPHMVS), James Cook University, Nguma-bada Campus, McGregor Rd., Cairns, QLD 4878, Australia; 3Australian Tropical Herbarium (ATH), James Cook University, Nguma-bada Campus, McGregor Rd., Cairns, QLD 4878, Australia; darren.crayn@jcu.edu.au

**Keywords:** wet tropics, climate change, metabolomics, tropical montane cloud forest, Wet Tropics World Heritage Area, plant secondary metabolites

## Abstract

The Australian Wet Tropics World Heritage Area (WTWHA) in northeast Queensland is home to approximately 18 percent of the nation’s total vascular plant species. Over the past century, human activity and industrial development have caused global climate changes, posing a severe and irreversible danger to the entire land-based ecosystem, and the WTWHA is no exception. The current average annual temperature of WTWHA in northeast Queensland is 24 °C. However, in the coming years (by 2030), the average annual temperature increase is estimated to be between 0.5 and 1.4 °C compared to the climate observed between 1986 and 2005. Looking further ahead to 2070, the anticipated temperature rise is projected to be between 1.0 and 3.2 °C, with the exact range depending on future emissions. We identified 84 plant species, endemic to tropical montane cloud forests (TMCF) within the WTWHA, which are already experiencing climate change threats. Some of these plants are used in herbal medicines. This study comprehensively reviewed the metabolomics studies conducted on these 84 plant species until now toward understanding their physiological and metabolomics responses to global climate change. This review also discusses the following: (i) recent developments in plant metabolomics studies that can be applied to study and better understand the interactions of wet tropics plants with climatic stress, (ii) medicinal plants and isolated phytochemicals with structural diversity, and (iii) reported biological activities of crude extracts and isolated compounds.

## 1. Introduction

Human activity and industrial development have led to significant and irreversible threats to the entire land-based ecosystem in the last century, primarily due to global climate changes. As the average global annual temperature continues to rise by 1 °C compared to the average temperature during the preindustrial era, experts have predicted that the temperature will further rise by 3–5 °C by the end of this century, owing to the increasing concentration of greenhouse gas (GHG), such as CO_2_ and methane, in the atmosphere [1,2]. Looking further ahead to 2070, the anticipated temperature rise ranges from 1.0 to 3.2 °C, based on the intensity of future GHG emission [3]. Climate change indirectly impacts species by diminishing the quantity and accessibility of habitat and eliminating species crucial for the survival of the species in question [4]. This impact is significantly felt in the tropical montane (TM) regions within the evergreen forests that are enveloped in persistent and frequent low-level clouds, forming unique ecosystems called TM cloud forests (TMCF) [5]. Indeed, recent studies, including climate model studies [6,7,8], predicted higher rates of temperature increase at higher elevations of TMCF regions than at lower elevations. Furthermore, alterations in reliability and quantity of precipitation in TMCF are anticipated due to reductions in cloud cover [5,9,10,11]. These factors would bring significant challenges to species in TMCF, leading to shifts in altitudinal ranges, reshuffling of species compositions, and increased risk of extinction [5].

The Australian TMCF, situated at or above 900 m in elevation (Figure 1) across the Wet Tropics World Heritage Area (WTWHA) of northeast Queensland, Australia, is no exception. Indeed, the impacts of climate change on Australian TMCF are anticipated to manifest within this century as the plant species are particularly vulnerable to climatic stress compared to the lowland species [7,12]. The WTWHA, which is considered the sixth most important protected area globally for conserving biodiversity [13], is home to over 3300 plant species, of which 700+ are endemic to the region, accounting for 18 percent of the nation’s total vascular plant species [14,15]. The ongoing project on climate-affected TMCF plants, led by the Australian Tropical Herbarium (ATH) at James Cook University, has identified 84 plant species vulnerable to climate change’s impact in the WTWHA. Some of these plants are used in traditional medicines, including Aboriginal bush medicines.

Plants affected by climate change or biotic stressors exhibit morphological and physiological plasticity to adapt, survive, and thrive [16]. Their adaptability relies on complex genetic or metabolic detection and communication systems that we are just starting to comprehend. Numerous research investigations have been carried out on various plants, such as *Arabidopsis*, aiming to unravel the intricate molecular mechanisms that plants exhibit in response to constantly changing environments [17,18,19,20,21]. More recently, technological advancements have enabled the acquisition of molecular data, including phenomics, epigenomics, transcriptomics, proteomics, and metabolomics data. These multidisciplinary approaches have enabled a better understanding of plants’ responses to various environmental changes linked to climate change, including drought and cold [22,23]. The plant produces various biomolecules, which serve different biological roles throughout plant life cycles. These biomolecules can be categorized under two major groups: primary metabolites (PMs) and secondary metabolites (SMs). Primary metabolites, such as proteins, sugars, and organic acids, are widely recognized for their role in essential plant physiological processes, including photosynthesis, photorespiration, and the tricarboxylic acid cycle [24,25]. Secondary metabolites, mainly phenolics, terpenoids, and alkaloids, are produced in response to competitive environmental factors for survival and fulfilling various physiological functions [26]. These SMs do not play a direct role in plants’ typical growth and survival; instead, they contribute to plant development and enhance resilience to stress [27]. For example, flavonoids are photo-protectants, shielding plants from damage caused by ultraviolet-B (UV-B) radiation [28,29]. Similarly, some terpenoids or alkaloids might act as antioxidants or osmotic regulators in response to abiotic stresses in plants [30,31]. In contrast, glucosinolates, limited to specific taxonomic groups, are considered to be essential antitoxins that play a crucial role in enabling plants to resist insect attacks [32].

Understanding the response patterns of these secondary metabolites to potential global climate changes is crucial due to their significance in plant growth, resistance, human health, and conservation efforts. Recently developed powerful omics techniques [33,34,35] and bioimaging [36] and biosensor tools [37,38] have been widely applied for understanding plant physiology, analysing the plant metabolome, and discovering novel metabolomic pathways in response to the changing environment [39]. However, there is no comprehensive review that examined recent metabolomic and phytochemical studies on climate-affected plants of Australian TMCF.

This review comprehensively analysed the literature on the available metabolomics studies conducted on the 84 TMCF plant species and discusses (i) recent developments in plant metabolomics studies that can be applied to study and understand the interactions of wet tropical plants with climatic stress, (ii) medicinal plants and isolated phytochemicals with structural diversity, and (iii) reported biological activities of crude extracts and isolated compounds. In doing so, we compiled and listed the plant species largely restricted to TMCF based on Hoyle et al. (2023) [5]. We compiled additional information from several sources: (i) records in the Atlas of Living Australia [40], (ii) the ‘Rainforest Key’ [41], and (iii) expert knowledge [42]. We searched for studies of metabolomic profiles, medicinal plants use, phytochemical contents, and biological activities in Google Scholar, MEDLINE Ovid, Scopus, PubMed, and journal websites using the following keywords: “wet tropics climate-affected plants”, “secondary metabolites in plants”, “metabolomics studies of plants”, “phytochemical analysis of plants affected by climatic impact”, “biological activities”, and accepted plant names and their synonyms. The information we collected was analysed and presented in tables and figures. Additionally, we utilized ChemDraw Professional software (v. 21.0.0) to create chemical structures, ensuring the accuracy of each structure by cross-referencing them with databases such as PubChem, ChemSpider, and HMDB databases.

## 2. Climate-Affected Australian Tropical Montane Cloud Forest Plants and Their Medicinal Uses

Using information from Hoyle et al. (2023) [5], expert opinion (botanists from the ATH at James Cook University in Cairns), and other literature, we found 84 climate-affected plant species largely restricted to Australian TMCF. The plant names were cross-checked using the Australian Plant Census [43], WFO Plant List [44], and Australian Tropical Rainforest Plants information system [41]. For these 84 plant species, we generated information on their botanical names, taxonomy, distribution, life form, and medicinal uses (Table 1). 

Of the 84 plant species, 54 are restricted to the WTWHA, 2 are endemic to TMCF within the WTWHA, and 4 are found outside Australia (Table 1 and Figure 2A). Of the 84 plant species, 29 were trees, followed by shrubs (28 species) and ferns (8 species) (Table 1 and Figure 2B). These 84 plant species belonged to 34 families, and the Orchidaceae family had the maximum number of species (8 species), trailed by the Ericaceae and Myrtaceae (6 species each) and Proteaceae and Rubiaceae (5 species each) (Figure 2C). Most of the families (15 families) had one species. When checked for their plant uses in traditional medicines, we found that most WTWHA plants were not used medicinally. This could be because most plants are endemic to WTWHA, and although they are used in Aboriginal bush medicines, these endemic species’ medicinal uses are not publicly available. Of 84 species, 43 belong to 29 medicinally important genera (Figure 2D). Of the 43 species, species of *Planchonella*, *Tasmania*, and *Litsea* were particularly indicated as traditionally used by Australian Aboriginal communities to treat various ailments, such as skin sores, scabies, and sore throat, as an antiseptic for boils, malaria, diarrhea, and cough (Table 1). Most of the genera were found to be used for medicinal purposes in traditional medicine systems of Asian countries, such as China, India, Indonesia, Malaysia, Japan, Taiwan, and Korea. 

A report from 2010 by the Commonwealth Scientific and Industrial Research Organization (CSIRO) [45] indicates that numerous tree species are at risk of experiencing mean temperatures that exceed their typical tolerance levels. To illustrate, the recent increase in temperature may already be placing stress on a lowland tree species that has adapted to thrive within a mean annual temperature range of 23.0 to 24.0 (measured at 200 masl). To survive in a comparable temperature environment, they must relocate more than 1000 m upward by 2080. However, habitat fragmentation will severely constrain their capacity to respond in this way [14]. Table 1 shows the conservation status of 84 Australian TMCF plant species affected by climate change. Of these, nearly half (41 species) are of conservation significance under Queensland State legislation: 21 species are listed as Vulnerable (V, sky blue bar), 6 are Near Threatened (NT, yellow bar), 3 are Endangered (E, light green bar), and 11 are Critically Endangered (CR, light red bar) (Table 1 and Figure 3). The remainder (43 species) are not currently listed as threatened, i.e., are categorized as Least Concern (LC), Special Least Concern (SL), or no conservation status indicated (No). 

Through modelling analysis, Costion et al. (2015) [7] projected significant declines in suitable habitat for 19 of the 84 TMCF plant species listed in Table 1, with estimates ranging from 17% to 100% by 2040 and at least 46% by 2080. Roeble (2018) [46] further refined these predictions, modelling 37 plant species (including 8 from Costion’s study) and predicting a mean habitat loss of 63% by 2085. The study predicted that 5 out of 37 modelled species (*Acrotriche baileyana*, *Gynochthodes constipata*, *Hymenophyllum whitei*, *Syzygium fratris*, *Tasmania* sp. *Mt. Bellenden Ker*) will experience a total loss of their suitable habitat by 2035 and another 2 species (*Cinnamomum propinquum* and *Leucopogon malayanus*) by 2085 [46]. A substantial increase in the suitable habitat through 2085 was only predicted for *Bubbia whiteana*. Overall, both studies [7,46] suggest that a significant portion of Australian TMCF plant species are either threatened or vulnerable to climatic stress. Hence, these plants must acclimatize and react swiftly to overcome environmental stresses or face extinction. Therefore, it is crucial to comprehend how plants react and adjust to shifts in their environment, striving to enhance their ability to withstand the challenges posed by climate change.

**Table 1 plants-13-01024-t001:** List of climate-affected Australian tropical montane cloud forest (TMCF) plants: their distribution, life form, conservation status and medicinal uses.

Botanical Name, Family, and Synonyms	Distribution	Life Form	Medicinal Uses	Metabolomics Profile Studied	Conservation Status (QLD)
Pteridophyta					
Dryopteridaceae					
*Parapolystichum grayi* (D.J.Jones) J.J.S. Gardner & Nagalingum Syn. *Lastreopsis grayi* D.L.Jones	Africa, the Neotropics, north-eastern Australia, Madagascar, Pacific Island, and southern Asia	Fern	NU	No	V
*Parapolystichum tinarooense* (Tindale) Labiak, Sundue & R.C.Moran Syn. *Lastreopsis tinarooensis* Tindale	Wet Tropics region (Australia)	Fern	NU	No	V
Hymenophyllaceae					
*Hymenophyllum whitei* Goy	Wet Tropics region (Australia)	Fern	NU	No	CR
Lindsaeaceae					
*Lindsaea terrae-reginae* K.U.Kramer	Wet Tropics region (Australia)	Fern	NU	No	E
Lycopodiaceae					
*Phlegmariurus creber* (Alderw.) A.R.Field & Bostock Syn. *Huperzia crebra* (Alderw.) Holub	Wet Tropics region (Australia), PNG, Hawaii	Epiphyte	*Phlegmariurus*/*Huperzia* species are traditionally used as vermifuge, purgative, and laxative [47].	No	CR
*Phlegmariurus delbrueckii* (Herter) A.R.Field & Bostock Syn. *Huperzia delbrueckii* (Herter) Holub	Wet Tropics region (Australia)	Epiphyte		No	V
Polypodiaceae					
*Oreogrammitis albosetosa* (F.M.Bailey) Parris Syn. *Polypodium albosetosum* F. M.Bailey	Wet Tropics region (Australia)	Fern	NU	No	V
*Oreogrammitis leonardii* (Parris) Parris Syn*. Grammitis leonardii* Parris	Wet Tropics region (Australia)	Fern	NU	No	V
*Oreogrammitis reinwardtii* Blume	Wet Tropics region (Australia), Sri Lanka, Philippines, Papua New Guinea, Solomon Islands, Malaysia	Fern	NU	No	V
*Oreogrammitis wurunuran* (Parris) Parris Syn. *Grammitis wurunuran* Parris	Wet Tropics region (Australia)	Fern	NU	No	SL
Magnoliophyta					
Apiaceae					
*Trachymene geraniifolia* F.M.Bailey	Wet Tropics region (Australia)	Herb	NU	No	NT
Apocynaceae					
*Parsonsia bartlensis* J.B.Williams	Wet Tropics region (Australia)	Climber	NU	No	V
Araliaceae					
*Hydrocotyle miranda* A.R.Bean & Henwood	Wet Tropics region (Australia)	Herb	*Hydrocotyle* species are used as anti-inflammatory herbs in Taiwanese folk medicines [48].	No	V
*Polyscias bellendenkerensis* (F.M.Bailey) Philipson	Wet Tropics region (Australia)	Shrub	*Polyscias* species are traditionally used to treat ailments, such as malaria, obesity, and mental disorders [49].	No	V
*Polyscias willmottii* (F.Muell.) Philipson	Wet Tropics region (Australia)	Tree		No	LC
Araucariaceae					
*Agathis atropurpurea* B.Hyland	Australia	Tree	*Agathis* species are traditionally used to treat myalgia and headaches [50].	Yes	LC
Arecaceae					
*Linospadix apetiolatus* Dowe & A.K.Irivine	Wet Tropics region (Australia)	Tree	NU	No	LC
Celastraceae					
*Hypsophila halleyana* F.Muell.	Wet Tropics region (Australia)	Shrub	NU	No	LC
Clusiaceae					
*Garcinia brassii* C.T.White	Wet Tropics region (Australia)	Tree	Infusions prepared from fruits of *Garcinia* species are traditionally used to treat dysentery, ulcers, and wounds [51].	No	LC
Cunoniaceae					
*Ceratopetalum corymbosum* C.T.White	Wet Tropics region (Australia)	Tree	NU	No	V
*Ceratopetalum hylandii* Rozefelds & R.W.Barnes	Wet Tropics region (Australia)	Tree	NU	No	LC
*Eucryphia wilkiei* B.Hyland	Wet Tropics region (Australia)	Shrub	NU	Yes	CR
Ebenaceae					
*Diospyros granitica* Jessup	Wet Tropics region (Australia)	Tree	*Diospyros* species are used traditionally used as sedative, astringent, carminative, febrifuge, anti-hypertensive, vermifuge, antidiuretic, and to relieve constipation [52].	No	NT
Elaeocarpaceae					
*Elaeocarpus linsmithii* Guymer	Wet Tropics region (Australia)	Tree	*Elaeocarpus* species are the source of popular spiritual beads (known as Rudraksha in Asia), which are used to treat various ailments, including mental/neurological disorders (stress, depression, anxiety, hypertension, epilepsy, migraine, and neuralgia), asthma, and also used as analgesic [53].	No	LC
*Elaeocarpus hylobroma* Y.Baba & Crayn	Wet Tropics region (Australia)	Tree		No	LC
Ericaceae					
*Acrotriche baileyana* (Domin) J.M.Powell	Wet Tropics region (Australia)	Shrub	NU	No	NT
*Dracophyllum sayeri* F.Muell	Wet Tropics region (Australia)	Tree	NU	No	V
*Leucopogon malayanus* subsp*. novoguineensis* (Sleumer) Pedley Syn. *Styphelia malayana* subsp*. novoguineensis* (Sleumer) Hislop, Crayn & Puente-Lel.	Wet Tropics region (Australia)	Shrub	NU	No	No
*Rhododendron lochiae* F.Muell. Syn. *Rhododendron notiale*, Craven	Wet Tropics region (Australia)	Shrub	*Rhododendron* species are used to prevent and treat many ailments, including respiratory disorders like asthma and bronchitis, dysentery, diarrhea, constipation, fever, cardiac disorders, and inflammation [54].	No	No
*Rhododendron viriosum* Craven	Wet Tropics region (Australia)	Tree		No	LC
*Trochocarpa bellendenkerensis* Domin	Wet Tropics region (Australia)	Tree	NU	No	LC
Escalloniaceae					
*Polyosma reducta* F.Muell.	Wet Tropics region (Australia)	Tree	NU	No	LC
Gesneriaceae					
*Boea kinneari* (F.Muell.) B.L.Burtt	Wet Tropics region (Australia)	Herb	NU	No	E
*Lenbrassia australiana* (C.T.White) G.W.Gillett	Wet Tropics region (Australia)	Shrub	NU	No	SL
Lamiaceae					
*Prostanthera albohirta* C.T.White	Mount Emerald, Wet Tropics region (Australia)	Shrub	Some *Prostanthera* species are used for topical applications to treat skin sores and infections [55,56].	No	CR
*Prostanthera athertoniana* B.J.Conn & T.C.Wilson	Wet Tropics region (Australia)	Shrub		No	CR
Lauraceae					
*Cinnamomum propinquum* F.M.Bailey	Wet Tropics region (Australia)	Tree	*Cinnamomum* species are most commonly used in traditional Chinese medicines to treat multiple disorders, including indigestion, microbial infections, and cough and cold [57].	Yes	V
*Cryptocarya bellendenkerana* B.Hyland	Wet Tropics region (Australia)	Tree	NU	Yes	LC
*Endiandra jonesii* B.Hyland	Wet Tropics region (Australia)	Tree	*Endiandra* species are traditionally used to treat rheumatism, headache, dysentery, pulmonary disorders, and uterine tumours [58].	No	V
*Litsea granitica* B.Hyland	Wet Tropics region (Australia)	Tree	*Litsea* species are used traditionally by Aboriginal communities to treat skin infections such as sores and scabies, and also used an antiseptic [59].	No	V
Myrtaceae					
*Leptospermum wooroonooran* F.M.Bailey	Wet Tropics region (Australia)	Tree	*Leptospermum* species are traditionally used in Malaysia to relieve menstrual and stomach disorders [60,61].	Yes	LC
*Micromyrtus delicata* A.R.Bean	Wet Tropics region (Australia)	Shrub	NU	No	E
*Pilidiostigma sessile* N.Snow	Wet Tropics region (Australia)	Shrub	NU	No	LC
*Rhodamnia longisepala* N.Snow & A.J.Ford	Wet Tropics region (Australia)	Shrub	*Rhodamnia* species are used traditionally in Indonesia to treat scars, toothache, and cough [62].	No	CR
*Syzygium fratris* Craven	Wet Tropic region (Australia)	Shrub	NU	No	CR
*Uromyrtus metrosideros* (F.M.Bailey) A.J.Scott	Wet Tropics region (Australia)	Shrub	NU	Yes	LC
Orchidaceae					
*Bulbophyllum lilianiae* Rendle	Wet Tropics region (Australia)	Epiphyte	*Bulbophyllum* species are traditionally used to treat skin diseases, cardiovascular diseases, and rheumatism [63].	No	LC
*Bulbophyllum wadsworthii* Dockrill Syn. *Oxysepala wadsworthii* (Dockrill) D.L.Jones & M.A.Clem.	Australia	Epiphyte		No	SL
*Bulbophyllum windsorense* B.Gray & D.L.Jones Syn. *Oxysepala windsorensis* (B.Gray & D.L.Jones) D.L.Jones & M.A.Clem.	Wet Tropics region (Australia)	Epiphyte		No	V
*Dendrobium brevicaudum* D.L.Jones & M.A.Clem. Syn. *Dockrillia brevicauda* (D.L.Jones & M.A.Clem.) M.A.Clem. & D.L.Jones	Wet Tropics region (Australia)	Herb, Epiphyte	*Dendrobium* species are used in traditional Chinese and Indian medicine systems as a source of tonic for longevity and also as an antipyretic, analgesic, astringent, and anti-inflammatory agent [64].	No	No
*Dendrobium carrii* Rupp & C.T.White Syn. *Australorchis carrii* (Rupp & C.T.White) D.L.Jones & M.A.Clem.	Wet Tropics region (Australia)	Herb, Epiphyte		No	SL
*Dendrobium finniganense* D.L.Jones Syn. *Thelychiton finniganensis* (D.L.Jones) M.A.Clem. & D.L.Jones	Wet Tropics region (Australia)	Herb, Epiphyte		No	SL
*Liparis fleckeri* Nicholls	Wet Tropics region (Australia)	Lithophyte	*Liparis* species are traditionally used in Chinese medicine to treat inflammatory diseases, including haemoptysis, metrorrhagia, traumatic haemorrhage, and pneumonia; they are also used to stop bleeding from wounds and to detoxify snakebite [65].	No	No
*Octarrhena pusilla* (F.M.Bailey) M.A.Clem. & D.L.Jones Syn. *Octarrhena pusilla* (F.M.Bailey) Dockrill	Wet Tropics region (Australia)	Epiphyte	NU	No	SL
Piperaceae					
*Peperomia hunteriana* P.I.Forst.	Wet Tropics region (Australia)	Herb	*Peperomia* species are traditionally used for treating pain and inflammation, gastric ulcers, asthma, and bacterial infections [66,67].	No	LC
Podocarpaceae					
*Prumnopitys ladei* (F.M.Bailey) de Laub Syn. *Stachycarpus ladei* (Bailey) Gaussen, *Podocarpus ladei* F.M.Bailey	Endemic to Wet Tropics Australia	Tree	Fruits and bark of *Prunmnopitys* species are considered medicinal [68].	Yes	No
Proteaceae					
*Austromuellera valida* B.Hyland	Endemic to Wet Tropics region	Tree	NU	No	V
*Helicia lewisensis* Foreman	Endemic to Wet Tropics region	Tree	*Helicia* species are used for treating mouth and skin sores and also kidney and gastric problems [59,69,70,71].	No	V
*Helicia recurva* Foreman	Endemic to Wet Tropics region	Tree		No	No
*Hollandaea porphyrocarpa* A.J.Ford & P.H.Weston Syn. *Hollandaea* sp. Pinnacle Rock Track (P.I.Forster PIF10714*)*	Endemic to Wet Tropics region	Shrub	NU	No	CR
*Nothorites megacarpus* (A.S.George & B.Hyland) P.H.Weston & A.R.Mast Syn. *Orites megacarpa* A.S.George & B.Hyland	Endemic to Wet Tropics region	Tree	NU	No	LC
Rubiaceae					
*Aidia gyropetala* A.J.Ford and Halford	Endemic to Wet Tropics region	Tree	*Aidia* species are used for treating body/muscle pains and pains due to gastric disorders [72].	No	LC
*Gynochthodes constipata* (Halford & A.J.Ford) Razafim. & B.Bremer Syn. *Morinda constipata* Halford & A.J.Ford	Endemic to Wet Tropics region	Climber	*Gynochthodes/Morinda* species are traditionally used for treating diabetes, inflammation, cancer, psychiatric disorders, and microbial infections [73].	No	LC
*Gynochthodes podistra* (Halford & A.J.Ford) Razafim. & B.Bremer Syn. *Morinda podistra* Halford & A.J.Ford	Endemic to Wet Tropics region	Climber		No	LC
*Ixora orophila* C.T.White Syn. *Psydrax montigena* S.T.Reynolds & R.J.F.Hend.	Endemic to Wet Tropics region	Shrub	*Ixora* species are used in Ayurvedic medicine against leucorrhoea, hypertension, menstrual irregularities, sprains, bronchitis fever, sores, chronic ulcers, scabies, and skin diseases [74].	No	No
*Wendlandia connata* C.T.White	Endemic to Wet Tropics region	Shrub	*Wendlandia* species are traditionally used for treating fever, dysentery, cough, hypertension, diabetes, constipation, inflammations, and hyperlipidemia [75].	No	NT
Rutaceae					
*Flindersia oppositifolia* (F.Muell.) T.G.Hartley & Jessup	Wet Tropics region (Australia)	Tree	NU	Yes	V
*Leionema ellipticum* Paul G. Wilson	Endemic to Wet Tropics region	Shrub	NU	Yes	V
*Zieria alata* Duretto & P.I.Forst.	Endemic to Wet Tropics region	Shrub	NU	No	CR
*Zieria madida* Duretto & P.I.Forst.	Endemic to Wet Tropics region	Shrub	NU	No	CR
Santalaceae					
*Korthalsella grayi* Barlow	Endemic to Wet Tropics region	Herb		No	LC
Sapindaceae					
*Mischocarpus montanus* C.T.White Syn. *Mischocarpus pyriformis* subsp*. retusus* (Radlk.) R.W.Ham, *Mischocarpus retusus* Radlk.	Wet Tropics region (Australia), New Guinea	Tree	NU	No	LC
Sapotaceae					
*Pleioluma singuliflora* (C.T.White & W.D.Francis) Swenson Syn. *Planchonella singuliflora* (C.T.White & W.D.Francis) P.Royen, *Pouteria singuliflora* (C.T.White & W.D.Francis) Baehni	Endemic to Wet Tropic region	Shrub	NU	No	LC
*Sersalisia sessiliflora* (C.T.White) Aubrév. Syn. *Pouteria sylvatica* Baehni, *Lucuma sessiliflora* C.T.White	Endemic to Wet Tropics region	Tree	NU	No	LC
*Planchonella* sp. Mt. Lewis (B.Hyland 14048) Qld Herbarium	Endemic to Wet Tropics region	Tree	*Planchonella* species have been used by Aboriginal medicine system to treat sores/sore throat and as an antiseptic for boils [59].	No	No
Solanaceae					
*Solanum dimorphispinum* C.T.White	Endemic to Wet Tropics region	Shrub	*Solanum* species have been traditionally used against infectious diseases and also as anti-microbial agents and insecticidal against mosquitoes [76].	No	LC
*Solanum eminens* A.R.Bean	Endemic to Wet Tropics region	Climber		No	LC
Symplocaceae					
*Symplocos bullata* Jessup Syn. *Symplocos* sp. North Mary (B. Gray 2543)	Endemic to Wet Tropics region	Shrub	*Symplocos* species are traditionally known for treating diseases such as malaria, ulcers, leprosy, leucorrhea, menorrhagia, and gynecological disorders [77].	No	LC
*Symplocos graniticola* Jessup	Endemic to Wet Tropics region	Shrub		No	V
*Symplocos oresbia* Jessup Syn. *Symplocos* sp. Mt Finnigan (L.J. Brass 20129)	Endemic to Wet Tropics region	Shrub		No	NT
*Symplocos wooroonooran* Jessup Syn. *Symplocos stawellii* var*. montana* C.T.White, *Symplocos cochinchinensis* var*. montana* (C.T.White) Noot	Endemic to Wet Tropics region	Shrub		No	NT
Thymelaeaceae					
*Phaleria biflora* (C.T.White) Herber Syn. *Oreodendron biflorum* C.T.White	Endemic to Wet Tropics region	Tree	*Phaleria* species are used for treating stomachache, general pain, diarrhea, lowering glucose/cholesterol levels in blood, and also known for anti-cancer properties [78].	No	V
Winteraceae					
*Bubbia whiteana* A.C.Sm. Syn. *Zygogynum semecarpoides* var*. whiteanum* Vink, *Bubbia semecarpoides* var*. whiteana* Vink	Endemic to Wet Tropics region	Shrub	NU	No	CR
*Tasmannia* sp. Mt Bellenden Ker (J.R.Clarkson 6571)	Wet Tropics region (Australia)	Shrub	*Tasmania* species are traditionally used for treating malaria, diarrhea, and cough [79].	No	LC

The scientific names and plant families follow the Australian Plant Census. Where taxonomy differs in “Plants of the World Online” [80], the synonym is given; distribution and plant life forms were sourced from the Atlas of Living Australia Field [41], the Australian Tropical Rainforest Plants system Field [42], and the Australian Tropical Rainforest Orchids [81]. Conservation status is as per the Queensland Nature Conservation Act 1992 [82]. Abbreviations—SL: Special Least Concern; LC: Least Concern; NT: Near Threatened; V: Vulnerable; E: Endangered; CR: Critically Endangered; No: Species for which no conservation status is indicated; NU: Not used medicinally.

**Figure 2 plants-13-01024-f002:**
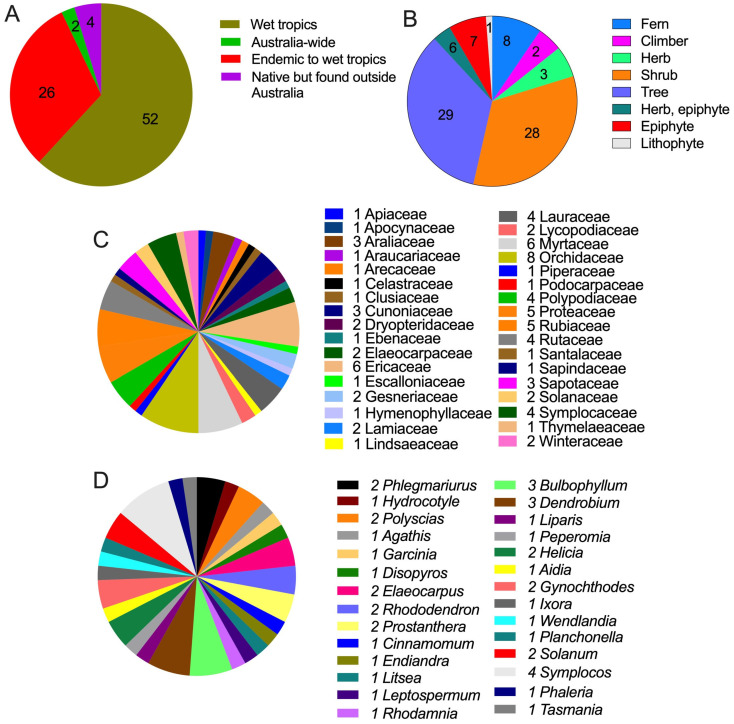
Climate-affected Australian tropical montane cloud forest (TMCF) plants in the Wet Tropics World Heritage Area (WTWHA), northeast Queensland: (**A**) distribution, (**B**) life form, (**C**) family diversity, and (**D**) medicinally important genus with species number. Distribution and plant life forms were sourced from the Atlas of Living Australia Field [41], the Australian Tropical Rainforest Plants system Field [42], and the Australian Tropical Rainforest Orchids [81]. Conservation status is as per the Queensland Nature Conservation Act 1992 [82].

**Figure 3 plants-13-01024-f003:**
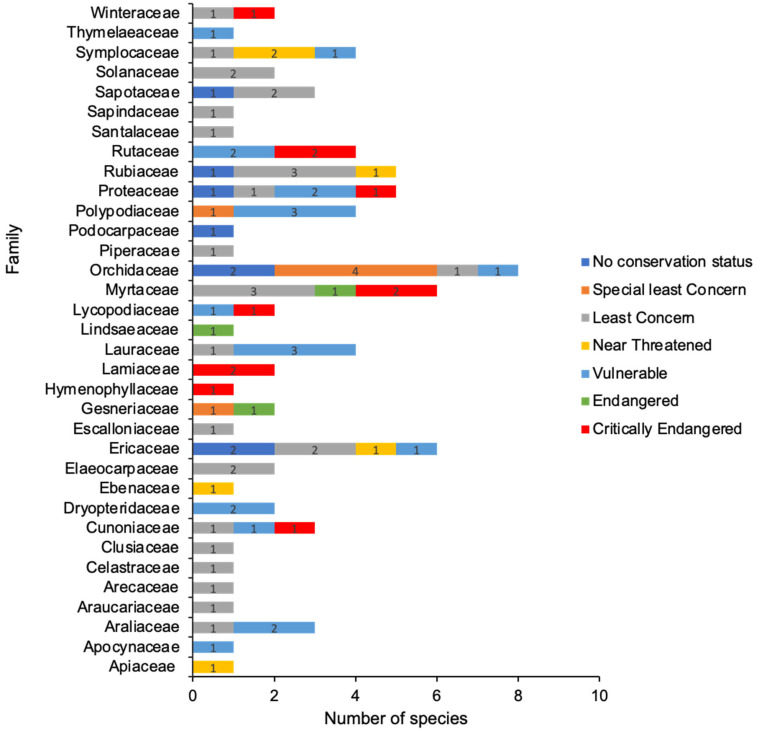
Conservation status of climate-affected Australian tropical montane cloud forest (TMCF) plants in the Wet Tropics World Heritage Area (WTWHA) in northeast Queensland. Conservation status is as per the Queensland Nature Conservation Act 1992 [82]. Different conservation status categories are represented by different colour codes, as shown in the figure legend, and numbers on bar plots represent plant species numbers.

## 3. Metabolomic Profile of Climate-Affected Plants in WTWHA

The anticipated impact of global climate changes on plant secondary metabolism is significant, but a comprehensive understanding of these effects is currently absent. Changes in the metabolome (defined as the complete set of metabolites found in a biological sample) can occur rapidly in seconds or minutes due to living organisms’ responses, acclimation, and adaptation to environmental conditions [83,84,85]. Investigations into climate effects on plants have shown that plants growing under various climatic stresses in their natural habitat produce various SMs that could potentially have a role in adaptation to the changing environment [86,87,88]. Studies have also revealed that abiotic stress factors, such as increased temperature and ultraviolet (UV) radiation, stimulate plants to reprogram their genetic codes for metabolic pathways, leading to the accumulation of new and unique secondary metabolites [89]. 

For example, it was demonstrated that elevated temperatures can lead to increased production of terpenoids, phenolic acids, and flavonoids in plants [90,91]. These compounds act as protective pigments when trees are exposed to UV-B radiation (wavelengths between 280 and 315 nm) [92]. Likewise, higher ozone (O_3_) concentrations have been linked to heightened production of antioxidant compounds such as glutathione, gamma-aminobutyric acid (GABA), terpenoids, and volatile organic compounds (VOCs) [93]. For example, the production of phenolics in plants plays an integral part in protecting mesophyll tissue from UV radiation and water stress [94,95]. It was discovered that drought conditions enhance plant productivity, leading to increased production of SMs, terpenes, complex phenols, and alkaloids [96,97,98]. Moreover, secondary metabolites with antioxidant properties, such as phenolic compounds and tocopherols, were known to scavenge the reactive oxygen species (ROS) generated, thus adapting to a new environment [99].

In addition to metabolites, plants store proteins like hydrolases, enzymes for detoxifying ROS, and enzymes for modifying cell walls. These proteins act as regulatory agents, governing plant growth and development [100]. Similarly, salinity also impacts the plant’s growth and development. It leads to an abnormal ion composition, causing toxicity, osmotic stress, producing ROS, cellular harm, and degrading membrane lipids, proteins, and nucleic acids [99]. In response to saline soil stress, plants undergo a biochemical process that produces ions that can act against ion toxicity and abnormal osmotic pressure developed from salinity [101]. In addition, when a huge amount of sodium (Na^+^) ions prevails, plants respond to an abundance of Na^+^ ions by activating a sophisticated defence system, which enables them to regulate cellular and ion balance effectively [102].

These studies enable us to understand the metabolite/micronutrient change patterns, including compositions, variations, and biosynthetic pathways resulting from plants’ responses to biotic and abiotic stressors, collectively known as plant metabolomics [103,104]. It can provide insights into plant phenotypic relations to their physiological and resistance development and biodiversity [105]. More than 200,000 secondary metabolites (SMs) have been identified [106] from over 391,000 plant species known worldwide [107] through metabolomics studies. The projected number for the plant kingdom is expected to surpass 200,000 [108,109]. Hence, plant metabolomics poses a significant hurdle for researchers in plant science. The comprehensive research workflow in plant metabolomics encompasses experimental planning, sample gathering, sample handling, sample preparation, detection and examination, data handling, as well as the analysis of metabolic pathways and networks [110]. 

From our literature review on 84 plant species affected by climate change in WTWHA of northeast Queensland, only nine species were studied for their metabolomic profiles/phytochemical contents (Table 2). A total of 279 metabolites (251 identified and 18 isolated) were identified/isolated from parts of 9 plant species. The identified metabolites were mostly flavonoids, terpenoids, alkaloids, and glucosides (Table 2). However, none of these metabolomics studies included in Table 2 were conducted to investigate their response to climatic stress conditions under in situ or ex situ conditions. There is a need for this type of study, but the challenge would be to control various factors influencing plant responses, which is why we see most of the studies conducted under controlled conditions in glass houses. Currently, our group is conducting a first-of-its-kind study on selected WTWHA plants, in which we are comparing the metabolome profile and chemical variation between the wild and the domesticated plant population.

## 4. Metabolomics Approaches, Tools, and Techniques Used in Plant Metabolomics

Understanding plants’ physiological and metabolomic responses to global change is key to identifying potential traits, including their genetic mutations and changes in their metabolomic pathways. Additionally, it is possible to predict potential changes in the composition of plant communities by assessing the ability of various plant species to adapt to environmental shifts [147,148]. Metabolites from living matters can be identified using (i) isolation techniques and (ii) metabolomics platforms. Metabolomics platforms, in general, rely on mass spectrometry (MS)-based techniques, namely capillary electrophoresis mass spectrometry (CE-MS), liquid chromatography-mass spectrometry (LC-MS), and gas chromatography-mass spectrometry (GC-MS) [149,150]. We found that GC-MS is the most used technique among the studies that analysed metabolites of Australian TMCF plants. Out of nine plants studied for metabolomics included in this review, metabolites from seven species were analysed using GC-MS (Table 2). Two innovative technological methods that do not require the use of metabolite chromatography include Nuclear Magnetic Resonance (NMR) analysis of unrefined extracts and the direct inspection of unrefined extracts using mass spectrometry (MS), specifically either quadrupole (Q) TOF-MS or ultra-high-resolution Fourier transform ion cyclotron MS (FT-MS) [105], which are discussed in-depth in later sections. Compared to conventional methods in the postgenomic era, metabolomics analysis offers numerous advantages and potential applications [150]. It has various steps, as shown in Figure 4, including sample preparation, spectra processing, data analysis, and metabolite identification. 

The NMR-based metabolomic analysis offers a potent, non-invasive method, delivering precise structural details about metabolites [151]. While metabolomic analysis using mass spectrometry is inherently destructive, it is highly sensitive and can detect traces of metabolites, and thus, it has gained more popularity [152]. Mass spectrometry (MS) methods are frequently integrated with chromatographic separation methods, including gas chromatography (GC) and liquid chromatography (LC) [152]. Only one metabolomics study applied LC-MS and NMR techniques to analyse the alkaloid diversity in the leaves of Australian *Flindersia* species, including *F. oppositifolia* (Table 2). Since the metabolome is a complex mixture of many small molecules, chromatographic separation is necessary prior to ion detection, particularly to distinguish isobaric compounds with a similar mass. Alternatively, the direct-infusion mass spectrometry (DIMS) approach is applied to measure metabolites directly without a prior chromatographic separation [153]. However, none of the studies that analysed Australian TMCF plants have applied either of these techniques. Most of the studies on plant-based metabolomics published so far have used the Orbitrap or TOF (time-of-flight) equipment [154]. One of the main reasons for using TOF equipment could be due to its mass resolution values (i.e., 30,000–40,000) [155,156], and the resolution power is unaffected by chromatography acquisition rates [157,158,159]. On the contrary, Orbitrap mass spectrometers can rapidly acquire tandem MS spectra up to 240,000 mass resolution, and thus, they are mainly applied in the shotgun metabolomics [157,160,161]. 

The DIMS methodology has recently been expanded to swift, high-throughput fingerprinting techniques employing advanced mass spectrometers with high resolution, such as Fourier transform ion cyclotron resonance (FT-ICR) mass spectrometers (FT-ICR-MS) [157]. Its extensive usage in plant metabolomics greatly helped understand plant development, responses to biotic and abiotic stresses, and exploring novel natural nutraceutical compounds [154,162]. The FT-ICR-MS has a higher resolution power (10^5^ to >10^6^), mass accuracy (typically 0.1–2 ppm), and sensitivity [163,164]. For instance, FT-ICR-MS can analyse and evaluate approximately 50,000 molecular formulas in complex samples, such as plant-derived crude essential oils [165,166]. Another reason for more usage of FT-ICR-MS is that this instrument has a wide range of ionisation sources, including electron spray ionisation (ESI), atmospheric pressure chemical ionization, and photoionization, thus enabling analyses of different sample types [167]. For example, Shahbazya et al. (2020) [168] used FT-ICR-MS to study the response of thyme plants (*Thymus vulgaris*) to drought stress. The study identified galactose metabolism as the most significant factor in drought adaptive response in thyme. Other studies, including metabolomics changes in poplar species in response to salinity stress [169] and UV-B radiation, also applied the same technique [170].

Nevertheless, because metabolites exhibit various chemical properties and are found abundantly in various cells, no single analytical platform can encompass the entire metabolome. Therefore, multi-omics technology has enabled the exploration of genes and metabolites in response to various climatic stress factors, particularly by combining transcriptomics and metabolomics approaches. For instance, Wang et al. (2021) [23], in their study about *Poa crymophila*, applied transcriptomics and metabolomics and identified the phenylpropanoid pathway as the main mechanism that facilitates this plant to survive in the unfavourable environment of Qinghai-Tibet Plateau. Liu et al. (2021) [22] also applied a combination of transcriptomics with physiological analyses to understand the chilling response in pumpkins and found that α-linolenic acid biosynthesis was one of the key pathways in the response.

These instruments employ three approaches to characterize metabolites, namely, (i) targeted analysis, (ii) untargeted analysis, and (iii) metabolic fingerprinting [171]. Unlike the targeted approach (identify a set of targeted metabolites with reference to available standards), the untargeted approach generates a large volume and complex data requiring specialised computational methods, such as artificial intelligence (AI) and machine learning (ML) algorithms, to process and interpret data [171]. In contrast, metabolic fingerprinting or exometabolomics involves characterising extracellular metabolites (i.e., metabolic by-products of organisms produced in response to environmental factors in which they survive) [172,173]. For plant metabolomic profiling, “ecometabolomics” is a commonly applied technique. The term “ecometabolomics” first appeared in the scientific literature in 2009 [33,174]. This study investigates how living organisms respond, acclimate, and adapt to environmental conditions by a nontargeted approach [83,84,85]. Metabolite identification can be achieved at four metabolite standard initiative (MSI) levels. Metabolite standard initiative level-1 (MSI-1) is considered the highest level of identification as it identifies metabolites after comparing with their chemical standards [175,176]. Level-2 (MSI-2) and level-3 identifications are only putative, as metabolites (MSI-2) or metabolite class (MSI-3) are not compared to their chemical standards, whereas MSI level-4 (MSI-4) putatively annotates unknown metabolites [175,176]. 

**Figure 4 plants-13-01024-f004:**
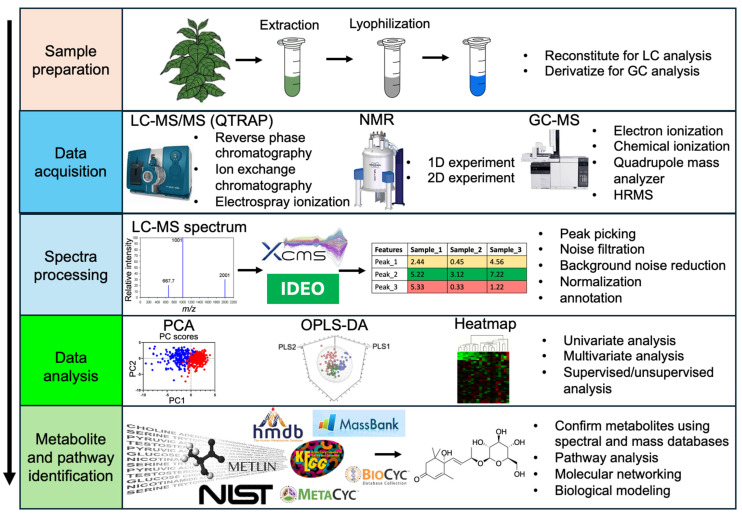
Common metabolomic workflow applied in plant metabolomics studies. The figure was adapted from Xu and Fu [177], and all databases’ logos used in this figure were obtained from their respective websites. Abbreviations—LC-MS: Liquid chromatography-mass spectrometry; NMR: Nuclear Magnetic Resonance; 1D: one-dimensional; 2D: two-dimensional; GC-MS: Gas chromatography-mass spectrometry; QTRAP: The Quadrupole Ion Trap; XCMS: eXtensible Computational Mass Spectrometry; HMDB: Human metabolome database; PCA: principal component analysis; OPLS-DA: Orthogonal Partial Least Squares Discriminant Analysis; NIST: National Institute of Standards and Technology; METLIN: Metabolite and chemical entity database.

## 5. Phytochemicals Isolated from Climate-Affected Plants in WTWHA

Out of 84 plant species included in this review, phytochemicals were isolated from only 3 plant species, namely, *Uromyrtus metrosideros*, *Flindersia oppositifolia*, and *Leionema ellipticum* (Table 2). A total of 19 compounds/secondary metabolites were isolated from these 3 plant species (Table 2), and these compounds belong to 4 different chemical groups (alkaloids, flavonoids, benzopyrans, and glucosides). For example, two new galloyl glucosides (galloyl-lawsoniaside A and uromyrtoside) and four known compounds were isolated from *Uromyrtus metrosideros* [132]. These six compounds were characterised using low- and high-resolution mass spectrometry (L/HRMS) and Nuclear Magnetic Resonance (NMR) spectroscopy. All three studies involving three plants were conducted to identify pharmacological drug leads (*U. metrosideros* and *F. oppositifolia*) and solve the taxonomic discrepancies (*L. ellipticum*). They did not suggest their role in response to climatic stress factors. 

## 6. Pharmacological Activities of Isolated Phytochemicals of Climate-Affected Plants in WTWHA

Studies have suggested that SMs, which function as plant defence mechanisms, possess intriguing pharmacological properties, including antioxidant and anti-inflammatory properties [178]. For instance, a novel galloyl-lawsoniaside A isolated from *U. metrosideros* leaf significantly suppressed pro-inflammatory cytokines, such as interferon-gamma and interleukins-17 (IL-17) and IL-18, and thus was identified as a new anti-inflammatory drug-lead molecule [132]. Osthol isolated from *Leionema ellipticum* also showed anti-inflammatory activity [143,146]. A study by Yeshi et al. [178] analysed crude extracts from the leaves of seven plant species endemic to WTWHA of FNQ. Five of the seven plant species showed potent antioxidant and anti-inflammatory activities in in vitro human peripheral blood cells (PBMCs) assay [178]. About 30 plant species growing in the WTWHA were reported as medicinal plants used for many years by indigenous communities to treat various diseases and ailments, including inflammation-related diseases [179]. Many metabolites identified through metabolomic studies (Table 2) were also studied for numerous biological properties (Table 2). Some major and bioactive metabolites were α-pinene, *p*-cymene, β-endemol, limonene, viridiflorene, E-β-farnesene, copaene, and β-caryophyllene (Table 2). Figure 5 shows some interesting structures of these isolated compounds. They showed a wide array of pharmacological activities, from anti-microbial to anti-cancer and anti-plasmodial properties. Of nine plant species listed in Table 2, four were tested for anti-inflammatory and anti-cancer properties, three each were tested for anti-microbial and antioxidant activities, two were tested for anti-allergic reactions, and the rest were tested for anti-diabetic, anti-malarial, and neuro-protective properties (one plant species each). For instance, galloyl-lawsoniaside A isolated from *U. metrosideros* leaf showed promising anti-inflammatory activity through significant suppression of pro-inflammatory cytokines, interferon-gamma (IFN-γ), and interleukin-17A (IL-17A) by phorbol myristate acetate/ionomycin (P/I)-activated cells [132]. Moreover, it also significantly suppressed the release of IL-8 by the anti-CD3/anti-CD28-activated cells [132]. There are increasing studies on identifying anti-inflammatory molecules by targeting the 5-lipoxygenase (5-LOX) pathway, as 5-LOX drives inflammation by producing inflammatory mediators, such as leukotrienes [180,181]. Osthol isolated from aerial parts of *Leionema ellipticum* showed selective inhibition of the 5-LOX pathway [143]. The anti-cancer/anti-tumour activity was mainly tested with crude extracts or essential oils by studying their inhibitory effect on tumour growth using tumour cell lines such as sarcoma 180 ascites tumour cells [125]. The anti-cancer activity was attributed to the major metabolite constituents, such as limonene, *p*-cymene, α-pinene, and viridiflorene (Table 2), and was not tested against the single compound. A few isolated compounds also exhibited anti-plasmodial activity. For example, pimentelamine C, isolated from the leaf of *Flindersia pimentaliana*, showed moderate anti-plasmodial activity against *Plasmodium falciparum* with IC_50_ values of 3.6 ± 0.7 (against chloroquine-sensitive strain) and 2.7 ± 0.3 (against chloroquine-resistant strains) [141]. 

## 7. Conclusions and Future Directions

The Australian tropical montane cloud forest (TMCF), which lies in the Wet Tropics World Heritage Area in FNQ, has rich and unique biodiversity, with over 700 endemic plant species. The current study identified that 84 plant species were affected by climate change, with some species already being endangered in their natural habitat. Recent studies of 37 of these species predicted a total loss of suitable habitat for five species by 2035 and seven species by 2085 if greenhouse gas emission (e.g., CO_2_) into the atmosphere continues at the current speed. Recently, many powerful technologies have been developed, including omics techniques, bioimaging, and biosensor tools, which have been widely applied to understanding plants’ physiology and metabolome and discovering novel metabolomic pathways in response to global climate change. However, our literature review revealed that these 84 Australian TMCF plants were scarcely studied for their biomolecules, and that we understand little about their medicinal uses, chemical profiles, and biological functions. Of 84 species, 43 belong to 29 medicinally important genera with various medical properties, and only 7 species were studied for their metabolite compositions. There is an urgent need for enhanced metabolomics studies of these least-studied plants, given that they are at risk of significant habitat loss because of climate change. Additionally, it is urgent to understand and identify potential traits in these least-studied plants, including possible genetic mutations that may have led to the change in the pattern of secondary metabolite accumulation and their metabolomic pathways in the adaptive response to climatic stress factors. Such studies will produce more data to holistically understand the interactive effect of climate change on the growth and fitness of these plants. This, in turn, would enable us to predict the adaptive response of plants specific to future climatic conditions and, thus, design the appropriate conservation measures to rescue those already identified as endangered and nearly threatened plant species. 

Many metabolites reported from those plants that have already been studied have shown numerous pharmacological activities. Studies have also reported that plants produce defensive/protective secondary metabolites in response to climate change. Most of these defensive secondary metabolites are antioxidative/anti-inflammatory. Therefore, Australian TMCF plants also present an exciting avenue for discovering novel pharmaceutical leads. 

## Figures and Tables

**Figure 1 plants-13-01024-f001:**
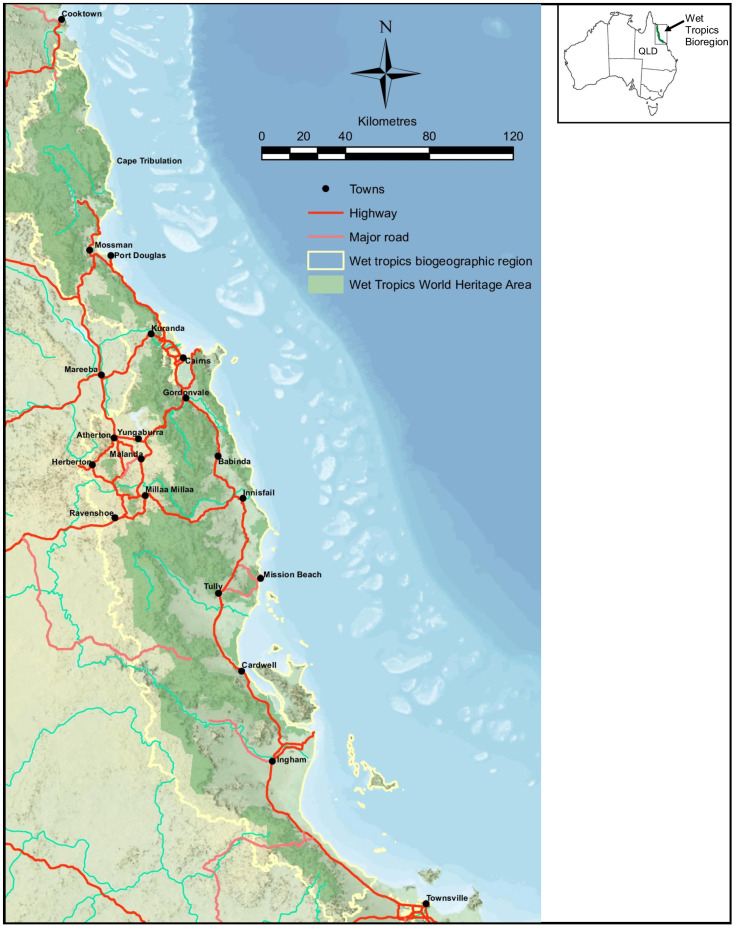
Map of Australia showing the State of Queensland and the Wet Tropics World Heritage Area (WTWHA) shaded in green. Tropical montane cloud forest (TMCF) is restricted to the mountaintops of the WTWHA, typically in areas above 900 m above sea level. A map depicting WTWHA was generated from the Wet Tropics Plan zoning map Edition 3.0 with the help of the Wet Tropics Management Authority office, Queensland.

**Figure 5 plants-13-01024-f005:**
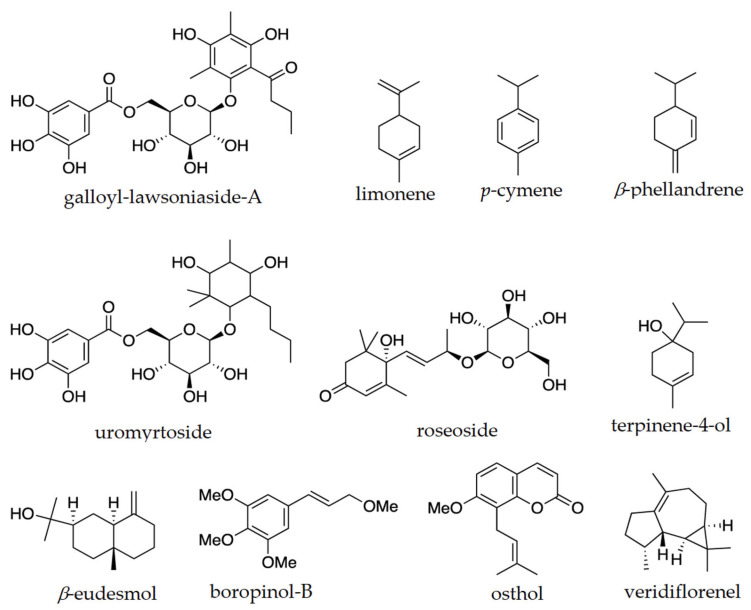
Chemical structure of bioactive compounds isolated/identified from the climate-affected Australian montane cloud forest (TMCF) plants (also used medicinally) in the Wet Tropics World Heritage Area (WTWHA), northeast Queensland.

**Table 2 plants-13-01024-t002:** List of climate-affected Australian tropical montane cloud forest (TMCF) plants studied for their phytochemical contents and bioactivity.

Botanical Name	Medicinal Uses	Number and Major Metabolites Identified	Isolated Compounds	Chemical Class	Biological Activities of Compounds
*Agathis atropurpurea*	*Agathis* species are traditionally used to treat myalgia and headaches [50].	27 metabolites; major metabolites are α-pinene, α-copaene, bicyclogermacrene, δ-cadinene, phyllocladane, and 16-kaurene [111]	NA	Terpenoid	Antimicrobial, antibacterial, antiviral, anti-cancer activity (α-pinene) [112,113,114], antioxidant activity (α-copaene) [115]
*Eucryphia wilkiei*	NU	2 unknown metabolites [116]	NA	Flavonoid	NA
*Cinnamomum propinquum*	*Cinnamomum* species are most commonly used in traditional Chinese medicines to treat multiple disorders, including indigestion, microbial infections, and cough and cold [57].	40 metabolites; Major metabolites are *p*-cymene, α-pinene, and β-eudesmol [117]	NA	Terpenoid	Anti-cancer activity (*p*-cymene) [118], anti-allergic and anti-angiogenic effect (β-eudesmol) [119,120]
*Cryptocarya bellendenkerana*	NU	39 metabolites; major metabolites are α-pinene, limonene, β-phellandrene, *p*-cymene, viridiflorene, *E*-β-farnesene, α-copaene, β-and α-selinene, δ-cadinene, bicyclogermacrene, calamenene, and cubeban-11-ol [121].	NA		Antioxidant, antidiabetic, anticancer, anti-inflammatory (limonene) [122,123], anti-fungal (β-phellandrene) [124], antioxidant and antitumour properties (viridiflorene) [125,126], insect repellent (*E*-β-farnesene) [127] antioxidant activity (copaene) [115].
*Leptospermum wooroonooran*	*Leptospermum* species are traditionally used in Malaysia to relieve menstrual and stomach disorders [60,61].	45 metabolites; major metabolitesare α-pinene, β-pinene, sabinene, α-terpinene, γ-terpinene, terpinen-4-ol and α-terpineol [128]	NA		Reduce skeletal muscle atrophy (sabinene) [129], antibacterial and antibiofilm activities (terpinene-4-ol) [130]
*Uromyrtus metrosideros*	NU	27 metabolites; major metabolites are α-pinene, β-pinene, spathulenol and aromadendrene [131]	norbergenin, bergenin, (*6S*,*9R*)-roseoside, (*4S*)-α-terpineol 8-*O*-β-D-(6-*O*-galloyl) glucopyranoside, galloyl-lawsoniaside A, and uromyrtoside [132]	Benzopyran, Glucoside,	Anti-inflammatory (galloyl-lawsoniaside A) [132]; reduced hypertension and allergic reaction (roseoside) [133,134]
*Prumnopitys ladei*	Fruits and bark of *Prunmnopitys* species are considered medicinal [68].	44-metabolites; major compounds are α-pinene, limonene, verbenone, and p-cymene. β-caryophyllene, caryophyllene oxide, spathulenol, and α-humulene [135]	NA		Antimicrobial, anticarcinogenic, anti-inflammatory, antioxidant, and local anesthetic effects (β-caryophyllene) [136,137,138]
*Flindersia oppositifolia*	NU	37 metabolites; major compounds are β-caryophyllene and bicyclogermacrene [139]; Identified 8 alkaloids from leaf [140].	pimentelamine A, pimentelamine B, pimentelamine C, 2-isoprenyl-*N*-*N*-dimethyltryptamine, 4-methylborreverine, borreverine, dimethylisoborreverine, quercitrin, and carpachromene [139]; harmalan, pimentelamine B, isoborreverine, skimmianine, kokusaginine, maculosidine, flindersiamine, 8-methoxy-*N*-methylflindersine [140].	Terpene, Alkaloid	Antiplasmodial (pimentelamine C) [141,142]
*Leionema ellipticum*	NU		3,4′,5-trimethoxyflavone-7-*O*-α-rhamnoside, boropinol-B, and osthol [143]	Flavonoid	Neuroprotective (boropinol-B) [144,145]; anti-inflammatory (osthol) [143,146]

Chemical class for isolated chemicals were referred from human metabolome database (https://hmdb.ca) (accessed on 10 January 2024); Abbreviations—NA: Not available; NU: Not used medicinally.

## Data Availability

Not Applicable.

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
