# Peer review of "Climate-Affected Australian Tropical Montane Cloud Forest Plants: Metabolomic Profiles, Isolated Phytochemicals, and Bioactivities"

_plants, 2024, doi:10.3390/plants13071024_

Round 1

Reviewer 1 Report

Comments and Suggestions for Authors

In this manuscript (plants-2930291) entitled "Climate-affected Australian Tropical Montane Cloud Forest Plants: Metabolomic Profiles, Isolated Phytochemicals, and Bioactivities" submitted to Plants, Ngawang Gempo and colleagues have comprehensively reviewed metabolomics studies conducted on these 84 plant species until now toward understanding their physiological and metabolomics responses to global climate change. In this manuscript, authors discussed recent developments in plant metabolomics studies that can be applied to study and understand the interactions of wet tropical plants with climatic stress, medicinal plants and isolated phytochemicals with structural diversity, and reported biological activities of crude extracts and isolated compounds. This review is interesting and well-written, but the current version of this manuscript is unsuitable for publication.

Major points:

1. I cannot find Figure 4 in this manuscript. In addition, Figure 5 should be placed in front of Figure 6 in the revision.

3. For the Figure 2, authors should consider to include citations in the revised legend.

4, For the section 6, Authors should extensively revise this section and rewrite this section in the frame of WTWHA plant research.

Minor points:

1. Full names of abbreviations like WTWHA and FNQ in the abstract should be spelt out at their first appearance. Authors should check all abbreviations employed in the manuscript.

2. Authors need to standardize references according to the Plants template.

Reviewer 2 Report

Comments and Suggestions for Authors

This study entitled "Climate-affected Australian Tropical Montane Cloud Forest Plants: Metabolomic Profiles, Isolated Phytochemicals, and Bioactivities" comprehensively reviewed metabolomics studies conducted on 84 plant species (Wet Tropics World Heritage Area of northeast Queensland, Australia) toward their physiological and metabolomics responses to global climate change. This review also analyzed and discusses: 1) recent developments in plant metabolomics studies that can be applied to study and understand the interactions of wet tropical plants with climatic stress, 2) medicinal plants and isolated phytochemicals with structural diversity, and 3) reported biological activities of crude extracts and isolated compounds.

 According to the Authors, the study would enable scientists to predict the adaptive response of plants specific to future climatic conditions and help them to design the appropriate conservation measures to rescue those already identified as endangered and nearly threatened plant species. Studies have also reported that plants produce defensive/protective secondary metabolites win response to climate change. Most of these defensive secondary metabolites are anti-oxidative/anti-inflammatory. Therefore, Australian TMCF plants also present an exciting avenue to discover novel pharmaceutical leads.

Please find my comments below:

 1.     Keywords: Please write like WTWHA (Wet Tropics World Heritage Area), etc; what is FNQ, elaborate. Keep all the key words in one place, otherwise it is hard to read.

 2.    Fig 3: Please elaborate Figure legends.

 3.   Page23, line 320: “Figure shows some interesting structures of these isolated compounds” repeated the sentence again on page 24, line 343. Remove all duplicate sentences like this one.

4.     Fig 5: What is the extraction solvent? Please include, if available. How did they reconstitute?

5.    Authors wrote that this review would enable us (1) to predict the adaptive response of plants specific to future climatic conditions and, thus, (2) design the appropriate conservation measures to rescue those already identified as endangered and nearly threatened plant species.

    How they will predict the adaptive response for future climatic condition? Please explain. What are those conservation measures to rescue? Please elaborate.

Round 2

Reviewer 1 Report

Comments and Suggestions for Authors

Authors have addressed my concerns in the revision.